# DiffCont: Continual Anomaly Detection with Diffusion Models and Outlier Rejection

## Abstract

We address continual anomaly detection in industrial inspection where new data flows in periodically in the form on new categories or newer types of defective samples. In such a setting, we would want the model to generalize to newer categories and shifting data distributions whilst maintaining its performance on previously learnt categories. Building on a diffusion-based U-Net with conditional generation and an auxiliary classifier, we introduce a Weibull-guided pipeline that serves two roles: (i) as a representative conditional generative distillation model, where a per-class Weibull model is fit on a projected embedding space and accepts only strong exemplar replay samples via threshold; (ii) as a per-class anomaly detection model, where the same heavy-tailed Weibull fit on the tail of normal embeddings converts distances into outlier probabilities for open-set anomaly detection. On MVTec-AD dataset split into 15 class-incremental experiences, our method achieves superior performance across experiences while presenting a light-weight anomaly detection workflow fit for industrial use.

## 1 Introduction

A visual system is subjected to constantly evolving data streams with a myriad of irregularities: categories appearing, tooling changes, sensors drift - and yet we expect yesterday's notion of "normal" to hold tomorrow. This is made incredibly clear in industrial inspection, represented well by MVTec-AD's 15 categories with normal-only training and diverse, previously unseen defects at test time. Here, adaptability is key to overcome catastrophic forgetting as new categories arrive Kirkpatrick et al. (2017). Naive fine-tuning is prone to such forgetting. Continual Learning (CL) is therefore necessary to trade plasticity for stability under realistic compute and memory budgets. Traditional CL approaches regularise (e.g. Elastic Weight Reconsolidation (EWC) Kirkpatrick et al. (2017)), replay exemplars (e.g., iCarl Rebuffi et al. (2017)) or synthesise data (e.g. generative replay Shin et al. (2017)), yet all of these fail under anomaly detection conditions as opposed to the more oft considered classification.

Generative replay is attractive in the continual anomaly detection setting, especially with recent diffusion models offering high fidelity, controllable generators Ho et al. (2020); Song et al. (2020b), as it avoids storing raw data. However, replay alone is not enough: generated samples can drift away from their true class manifolds, degrading both representation learning and the calibration of threshold scores for anomalies. At the same time, anomaly detection can be treated as an open-set problem Mundt et al. (2022). Combining these together, we obtain generative distillation with outlier rejection of out-of-distribution samples, thereby preserving the entire reverse process across experiences, making diffusion replay viable at scale. Even with this strengthened replay, anomaly detection can essentially be treated as Open-set recognition and its Extreme Value Theory (EVT) dictate that we should explicitly model the tails, so that the system can reject the unknown rather than force a closed-set guess.

We embrace an experience-centric view by coupling diffusion-based CL with explicit tail modelling on a per-class basis. More concretely, we train a diffusion U-Net guided by an auxiliary classifier and a 1x1-convolutional projector at the bottleneck to define a stable, low-dimensional embedding where experience-to-experience comparisons are meaningful. Within this space, we fit Weibull models for every class to the right tail of the normalised embeddings. These models play a dual role in the training loop. First, they screen samples generated for generative distillation - rejecting outliers

and ensuring that clean, representative samples are replayed. Secondly, at test time, the distances from the class centre are converted into outlier probabilities based on the CDF of the heavy-tailed Weibull distribution for that class which are then thresholded to classify the samples as 'normal' or 'anomaly'.

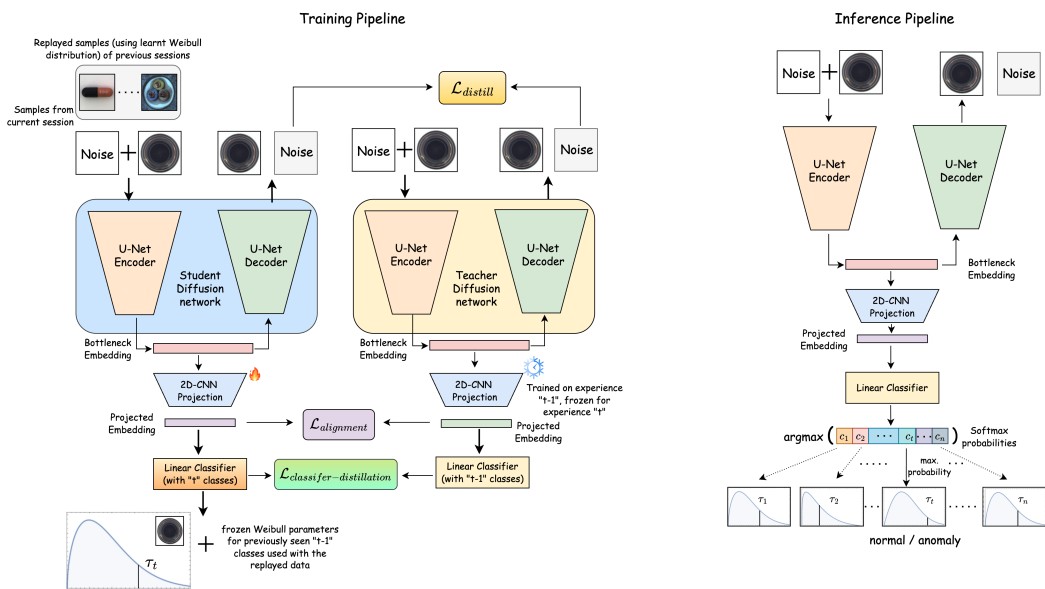

Figure 1: DiffCont Continual Anomaly Detection Workflow. First a noised image is passed to the diffusion U-Net. Secondly, a projected embedding space is trained on the first experience using a 1x1 Conv2D layer (this is later frozen). Then a linear classifier is trained on the output of the projection space, $P$ Next, we perform generative distillation between experiences using a typical teacher-student setup. In this scenario, the generated images must be be below a per-class Weibull threshold such that we are replaying representative samples between experiences. Finally, the samples are classified as normal or anomalous depending on the output of the CDF of a class-conditioned Weibull distribution being below a particular threshold, which is determined using a validation set. During inference, projected embeddings are extracted from the U-Net Bottleneck + Projector P and classified by the Classifier Head H into class $c$. We select the appropriate Weibull distribution (as was fit for class $c$ during training) to obtain the cumulative probability for the sample which is thresholded by $\tau_c$ into 'normal' or 'anomaly' class.

## 2 RELATED WORK

### 2.1 DIFFUSION MODELS

Diffusion models have become a cornerstone in generative modelling, especially for high-quality image synthesis. These models learn to gradually transforming noise into data samples through a denoising process, effectively modelling complex distributions. Denoising Diffusion Implicit Models (DDIM) improves sample quality and speed using non-Markovian diffusion processes Song et al. (2020a). Latent Diffusion Models (LDM) further accelerate generation by operating in a lower-dimensional latent space while preserving high fidelity Rombach et al. (2022). Model distillation techniques have been applied to diffusion models to reduce sampling steps, enabling faster and more efficient inference without significant quality loss Salimans & Ho (2022).

### 2.2 CONTINUAL LEARNING WITH REPLAY METHODS

Continual learning aims to mitigate catastrophic forgetting when models learn from a stream of tasks. Replay-based methods store and rehearse past examples to maintain performance. Experience Replay (ER) is a foundational approach that intermittently retrains on buffered samples from

previous tasks Rolnick et al. (2019) DER++ extends replay by combining rehearsal with knowledge distillation to align feature representations across tasks, yielding improved stability and plasticity Buzzega et al. (2020).

### 2.3 OUTLIER REJECTION

Ensuring robust model performance under distributional shifts in Continual Learning requires effective mechanisms for outlier rejection. Mundt et al. (2019) proposed a unified probabilistic framework for outlier rejection leveraging likelihoods under learned model distributions based on variational inference in a single deep autoencoder model. This formulation enables principled detection of novel or corrupted inputs by evaluating the joint likelihood of observations, surpassing heuristic or threshold-based anomaly rejection strategies. The approach constrains the approximate posterior by identifying high-density regions associated with correctly classified samples. These bounds serve a dual role: they facilitate reliable separation of unseen, out-of-distribution inputs from previously learned tasks, and they guide generative replay by restricting synthesis to strictly in-distribution data, thereby mitigating catastrophic interference.

### 2.4 ANOMALY DETECTION ON MVTEC-AD

The MVTec Anomaly Detection (AD) datasetBergmann et al. (2019a) constitutes a primary benchmark for unsupervised anomaly detection methods, particularly within industrial inspection contexts. It comprises of high-resolution images across 15 different objects, with pixel-wise annotations for a broad spectrum of defect types, making it invaluable for evaluating anomaly segmentation and classification algorithms. Contemporary methods evaluated on MVTec AD range from autoencoder-based reconstruction Yapp & Doan (2024); Cui et al. (2023), adversarial frameworks Akcay et al. (2018); Schlüter et al. (2022), and deep feature embeddings using pretrained vision models Roth et al. (2022); Zheng et al. (2022), to more recent approaches incorporating multiscale analysis and probabilistic scoring Ruff et al. (2018); Shin et al. (2023). Despite considerable advances, achieving generalizable anomaly detection performance across such diverse settings remains a central open challenge.

Continual learning in anomaly detection represents a rapidly evolving field, motivated by the need for robust systems that can adapt to evolving data distributions without catastrophic forgetting. In industrial and surveillance contexts, the nature of anomalies and normal patterns often changes over time, necessitating models capable of sequential adaptation while preserving learned knowledge from prior tasks.

## 3 METHODOLOGY

### 3.1 PROBLEM FORMULATION AND NOTATION

We study class-incremental anomaly detection on MVTec-AD, where each experience $e_j$ introduces a single category, where only the data from that experience is accessible at training time. Training uses only defect-free 'normal' images $X_{norm}^{train}$ of the current class, and evaluation mixes normal $\bigcup_{j=0}^{14} X_{\text{norm}, j=i}^{\text{test}}$ and anomalous images $X_{anomaly}^{test}$ for all seen classes with binary labels for anomaly detection.

Let $\mathcal{C}_t$ denote the set of classes observed up to and including experience $t$, and let $f_\theta(x, s, c)$ be a class-conditional diffusion model with de-noising step index $s \in \{0, \ldots, S\}$ and class label $c \in \mathcal{C}_t$. The model is augmented with a Projector $P$ that yields a whitened embedding $z(x) \in \mathbb{R}^d$ used for extreme value theory (EVT) calibration, where empirically $d = 2304$ due to the projection and spatial configuration at $32 \times 32$ resolution.

After each experience, per-class detectors are fit on distances $d_c(x) = \|z(x) - \mu_c\|_2^2$ computed over current-class embeddings, where $\mu_c$ is the class mean in the projected feature space. Test-time anomaly decisions use the model's built-in classifier to predict $\hat{c}$ and then apply the corresponding per-class detector to produce cumulative probability scores which are then thresholded to produce binary anomaly predictions.

## 3.2 OUTLIER REJECTION USING WEIBULL MODELS

Following Mundt's unified probabilistic framework for continual learning Mundt et al. (2022), we define acceptance regions in a shared latent space using EVT.

For each class $c$, a Weibull tail model is fit to the upper tail of $d_c(x)$ using libMR, inducing a per-class cumulative distribution:

$$s_c(d) = 1 - \exp\left(-(d/\lambda_c)^{k_c}\right)$$

This Weibull CDF score $s_c$ is later thresholded to classify samples as 'normal' or 'anomaly'.

## 3.3 GENERATION MEMORY BUDGET

We replace a persistent raw-image buffer with a regenerating, quality-gated generative buffer that is rebuilt after each experience. The system samples $K$ images per previously seen class from the frozen teacher and filters them via the Weibull-based acceptance statistic (Weibull CDF $\leq \tau_{\text{gen}} = 0.5$) to obtain a replay set of effective size $M = K \cdot |\mathcal{C}_{j-1}|$ interleaved during current-task optimization, where $K = 100$.

## 3.4 MODEL ARCHITECTURE: DISTILLATION, DIFFUSION AND WEIBULL MODELS

As we can see in Figure 1, the backbone is a UNet2D with a Projector P and a linear classifier head H, enabling joint diffusion training and class prediction for EVT fitting in a shared feature space at $32 \times 32$ resolution with Adam learning rate $2 \times 10^{-4}$.

We save the teacher model after each experience and use DDIM sampling for replay generation. The student model then performs knowledge distillation from the teacher on the replayed samples.

At each experience, we expand class embeddings as needed, construct a Weibull-gated generative replay set from the frozen teacher, train the student on interleaved current and replay data with denoising and classification losses while keeping the projection frozen, fit the current-class Weibull, and evaluate across all seen experiences.

## 3.5 TRAINING ALGORITHM

---

**Algorithm 1** Generative distillation with outlier rejection (Weibull)

---

**Initialize:** U-Net, Projector, Classifier for $\mathcal{C}_0 = \{c_0\}$ { Setup model for first class}
Train on $X_{norm,j=0}^{train}$, freeze Projector {Learn initial representations and fix the Projector P}
Fit Weibull $W_1(\tau_0, \kappa_0, \lambda_0)$ on projected embeddings {Learn parameters for the distribution}
**for** $t = 1, \ldots, 14$ **do**
   $\mathcal{C}_j \leftarrow \mathcal{C}_{j-1} \cup \{c_j\}$ {Add new class to set}
   Teacher $\leftarrow$ U-Net {Save current model}
   Expand U-Net, Classifier Head H to support $|\mathcal{C}_j|$ classes {Accommodate new class}
   **Generate replay:** $\mathcal{R}_t \leftarrow \emptyset$ {Initialize replay buffer}
   **for** $c \in \mathcal{C}_{j-1}$ **do**
     **repeat**
       $\tilde{x} \sim \text{DDIM}(\text{Teacher}, c)$ {Sample from frozen teacher}
       $z = \text{Projector}(\text{Teacher}(\tilde{x}))$ {Extract embedding}
       $s_c = W_c^{-1}(\|z - \mu_c\|_2)$ {Compute Weibull CDF score}
     **until** $s_c \leq \tau_{gen}$ and $|\mathcal{R}_j^{(c)}| = K$ {Accept if quality threshold met}
     $\mathcal{R}_j \leftarrow \mathcal{R}_j \cup \mathcal{R}_j^{(c)}$ {Add to replay set}
   **end for**
   **Train:** U-Net, Classifier on $X_{norm,t}^{train} \cup \mathcal{R}_j$ {Joint training on current + replay}
   Minimize $\mathcal{L}_{total}(\theta)$
   Keep Projector frozen {Preserve embedding geometry}
   **Fit EVT:** $\mu_j \leftarrow \mathbb{E}[\text{Projector}(\text{U-Net}(X_{norm,j}^{train})]$ {Compute class mean}
   $d_j \leftarrow \{\|\text{Projector}(\text{UNet}(x)) - \mu_t\|_2^2 : x \in X_{norm,j}^{train}\}$ {Distance to mean}
   $W_j \leftarrow \text{FitWeibull}(d_j, \text{tailsize})$
**end for**

---

# 4 EXPERIMENTAL SETUP

## 4.1 DATASET AND PROTOCOL

We evaluate on MVTec-AD Bergmann et al. (2019b) with 15 underlying classes making up the different objects represented in the data. Therefore, we adopt a class-incremental learning setup Masana et al. (2022) with 15 experiences, where each experience $e \in \{1, ..., 15\}$ introduces exactly one class and contains only normal images from that category for training. The MVTec class here refers to the different categories of industrial objects in the MVTec-AD dataset and the anomaly class refers to the image being classified as 'normal' or 'anomaly' for the anomaly detection task which is what most of the results capture.

## 4.2 DATA SPLITS PER CATEGORY

- Train (normal-only): Official MVTec-AD training split
- Test (normal + anomaly): Official test split uses normal and anomalous images

We do not use any anomalous images for training, fitting or classification.

## 4.3 BACKBONE

We use a diffusion U-Net with 1000 training steps. When running the U-Net, we randomly intialize a timestep $t$ and noise that will be added to the image. These are then passed to the U-Net which outputs the predicted noise which must be removed from the noisy image to de-noise it. We also use the U-Net for embeddings extraction which has been discussed later.

## 4.4 CLASSIFIER HEAD AND PROJECTED EMBEDDING SPACE

We attach a Projector $P$ at the U-Net bottleneck featuring a 1x1 Conv2D Layer ($256 \rightarrow 64$ channels) and flattening. P is only trained in the first experience and later frozen to define a canonical coordinate system. All embeddings below are taken after P. The model is aided by a linear classifier head H which classifies samples into MVTec classes.

## 4.5 WHITENING

We normalize every projected embedding P by taking mean and variance over P and apply whitening before any distance operations are applied: $\hat{z} := \frac{(z-\mu)}{\sigma}$.

The Projector P coupled with Whitening makes sure that we map all embeddings to a shared embedding space. Additional losses have been added to ensure that this latent space stays fixed across experiences.

## 4.6 WEIBULL FITTING (EVT)

For each underlying MVTec class, $c$, from the inlier distances, we take the top 10% right-tail and fit a 2-parameter Weibull $(k_c, \lambda_c)$ by MLE. All fitting is done in whitened space. At inference, we classify the sample using the classifier head H, pick the corresponding weibull model which had been fit during training and obtain the cumulative probability of the sample.

## 4.7 GENERATIVE REPLAY AND OUTLIER REJECTION

During training, we use generative replay to mitigate forgetting. At experience $e > 0$, the teacher (as a snapshot from $e - 1$) generates candidate clean $x_0$ samples conditioned on past classes. Each candidate is screened to have cumulative outlier probability less than a quality threshold $\tau$ in the projected whitened space. For the Weibull distribution corresponding to the class, it is accepted if it is below the Weibull quality threshold $\tau$ and rejected otherwise. Only accepted generations enter the replay buffer and are then used for generative distillation.

## 4.8 TRAINING LOSSES

- Diffusion Loss: Standard MSE between added noise ($\epsilon$) and predicted noise ($\epsilon_\theta(\mathbf{x}_t, t)$) at a random t

$$\mathcal{L}_{\text{diff}}(\theta) = \mathbb{E}_{t,\mathbf{x}_0,\boldsymbol{\epsilon}} \left[ \|\boldsymbol{\epsilon} - \boldsymbol{\epsilon}_\theta(\mathbf{x}_t, t)\|_2^2 \right] \tag{1}$$

- Classifier Loss: Cross-entropy loss for classification

$$\mathcal{L}_{cls}(\theta) = -\sum_{i=1}^{N} \sum_{c=1}^{C} y_{i,c} \log(\text{softmax}(f_\theta(\mathbf{h}_i))_c) \tag{2}$$

- Teacher Knowledge Distillation: MSE Between student and teacher noise predictions at the same $(x_t, t)$

$$\mathcal{L}_{KD}(\theta) = \mathbb{E}_{t,\mathbf{x}_0,\boldsymbol{\epsilon}} \left[ \|\boldsymbol{\epsilon}_{\theta_s}(\mathbf{x}_t, t) - \boldsymbol{\epsilon}_{\theta_t}(\mathbf{x}_t, t)\|_2^2 \right] \tag{3}$$

- Embedding Alignment Loss - L2 norm between student and teacher projected embeddings

$$\mathcal{L}_{align}(\theta) = \mathbb{E}_{t,\mathbf{x}_0,\boldsymbol{\epsilon}} \left[ \|\mathbf{z}_{\theta_s}(\mathbf{x}_t, t) - \mathbf{z}_{\theta_t}(\mathbf{x}_t, t)\|_2^2 \right] \tag{4}$$

- Center Loss - L2 norm between projected embedding and batch mean to ensure that the embeddings of a given class map to the similar vicinty in the embedding space

$$\mathcal{L}_{center}(\theta) = \frac{1}{N} \sum_{i=1}^{N} \|\mathbf{z}_i - \boldsymbol{\mu}_{\text{batch}}\|_2^2 \tag{5}$$

- Classifier Distillation Loss - KL divergence between teacher and student classifier head output classification probability distribution

$$\mathcal{L}_{cl\_distill}(\theta) = \text{KL}\left(P_{\theta_t}(\mathbf{e}) \| P_{\theta_s}(\mathbf{e})\right) \tag{6}$$

$$\begin{aligned} \mathcal{L}_{total}(\theta) = \mathcal{L}_{diff} + w_{cls} * \mathcal{L}_{cls} + w_{center} * \mathcal{L}_{center} \\ + w_{cl\_distill} * \mathcal{L}_{cl\_distill} + w_{KD} * \mathcal{L}_{KD} \\ + w_{align} * \mathcal{L}_{align} \end{aligned} \tag{7}$$

## 4.9 EVALUATION

We report per-experience and average Image-level AUPR, Accuracy and f1-score. Table 1 notes how these values vary across experiences. Table 2 compares our results with those from other baselines for this task.

## 4.10 ABLATIONS

- 4.1 Ablation on Training Epochs: We compare the 1 epoch results with the 100 epoch results
- 4.2 Ablation on Tailsize: We ablate on the percent of the train data that is considered as the tail while fitting the Weibull Distribution. We compare having 5% of the data as tail results with the 10% results

## 5 RESULTS

**Main Results**

Table 1: Performance metrics across continual learning experiences

| Experience | AUPR | Accuracy | F1 Score |
|---|---|---|---|
| 0 | 0.9673 | 0.7108 | 0.7736 |
| 1 | 0.7088 | 0.5322 | 0.5477 |
| 2 | 0.7233 | 0.7233 | 0.8394 |
| 3 | 0.7324 | 0.7324 | 0.8455 |
| 4 | 0.7996 | 0.6143 | 0.7333 |
| 5 | 0.7164 | 0.7164 | 0.8348 |
| 6 | 0.7204 | 0.7204 | 0.8375 |
| 7 | 0.7316 | 0.7316 | 0.8450 |
| 8 | 0.7491 | 0.7491 | 0.8565 |
| 9 | 0.7473 | 0.7395 | 0.8497 |
| 10 | 0.7458 | 0.7458 | 0.8544 |
| 11 | 0.7465 | 0.7391 | 0.8485 |
| 12 | 0.7213 | 0.7217 | 0.8384 |
| 13 | 0.7236 | 0.7236 | 0.8397 |
| 14 | 0.7293 | 0.7293 | 0.8434 |

**Comparison with baselines**

Table 2: F1-score comparison across anomaly detection models. **Best** results are bolded and worst results are underlined

| Metric | DRÆM | PatchCore | PaDiM | CFA | STPFM | EfficientAD | FastFlow | Diff-Cont (Our) |
|---|---|---|---|---|---|---|---|---|
| F1-Score | 0.87 | **0.97** | 0.83 | 0.94 | 0.89 | 0.89 | 0.89 | 0.84 |

**Ablation Studies: On Epochs**

Table 3: Performance metrics across training epochs

| Epochs | AUPR | Accuracy | F1 Score |
|---|---|---|---|
| 1 | 0.7293 | 0.7293 | 0.8434 |
| 100 | 0.7481 | 0.6928 | 0.8066 |

5.4 Ablation Studies: On Tailsize

Table 4: Performance metrics at different training completion percentages

| Tailsize % | AUPR | Accuracy | F1 Score |
|---|---|---|---|
| 5% | 0.724 | 0.682 | 0.807 |
| 10% | 0.7293 | 0.7293 | 0.8434 |

# 6 DISCUSSION

## 6.1 TRADEOFFS AND LIMITATIONS

Accuracy and f1-score can be seen to follow a similar trend wherein the scores gradually increase while the AUPR can be seen to decrease and flat-line around 70-75%. Our method exhibits competitive performance against baselines while maintaining a light-weight pipeline for industrial applications. This proves that the model is able to get better over time, while not catastrophically forgetting previous classes, which is crucial in the industry where data flows in large volumes. Our ablations show that training the fitting of the Weibull with a slightly higher tailsize percentage helps capture information about the distribution more accurately.

Despite our best efforts, there still appears to be room for improvement in our method, and we hypothesise that this could be achieved through ensuring the representations that the UNet learns for

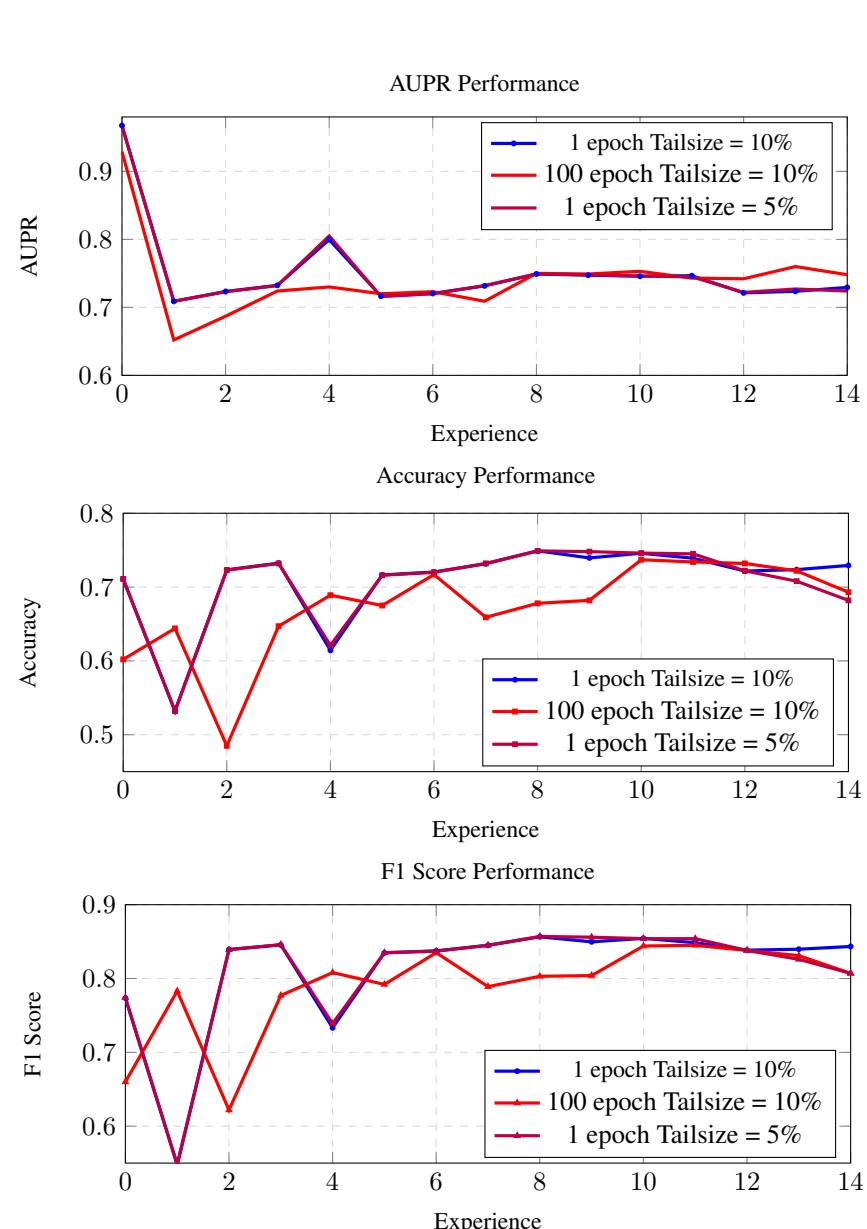

Figure 2: Variation of Performance Metrics across experiences

generative replay are of sufficiently high quality such that the replayed samples can be used in future experiences to tune the rejection thresholds for the per-class Weibull's. Further to this, we believe that the projection space that the model uses could be improved to ensure rotational-invariance and meaningful embeddings which naturally distinguish between normal and anomalous samples on a per-class basis. Without proper representations with the correct amount of overlap between classes, the embedding space becomes either a lookup table (with too little overlap) or loses all meaning altogether (with too much). This could be

**Why we chose U-Net bottleneck embeddings** During the forward pass, we pass the noised images along with the added noise to the model. As a result, the bottleneck embedding can be seen not only as a representation of the image but also of the reconstruction error associated with the sample. We would expect the U-Net to take more epochs to train so that it can generate better quality images. However, because the Weibull distribution is fit on the projected embeddings, which capture reconstruction loss, we are able to achieve superior performance with less training.

## 7 CONCLUSION

We propose a light-weight, easy-to-train anomaly detection workflow for industrial applications using a combination of generative distillation, Diffusion-based methods and outlier rejection through Weibull models. On the MVTec-AD dataset, we demonstrate how our method achieves competitive results with baselines when comparing their f1-scores. The use of deep generative models allows us to perform continual learning without storing a replay buffer which makes our method attractive for industrial applications. We are optimistic that this will inspire further research into the overlap between Continual Learning, Diffusion models and Extreme Value Theory techniques - as we believe this to be a mathematically grounded approach to anomaly detection, leveraging the positive aspects of each respective field.

## DECLARATION ON GENERATIVE AI

During the preparation of this work, the author(s) used OpenAI ChatGPT-4o, ChatGPT-5 and Claude Sonnet 4.0 in order to: Grammar and spelling check, Paraphrase and reword. After using these tool(s)/service(s), the author(s) reviewed and edited the content as needed and take(s) full responsibility for the publication's content.

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
