# OpenReview forum: "DiffCont: Continual Anomaly Detection with Diffusion Models and Outlier Rejection"
_ICLR.cc/2026/Conference — ICLR 2026 Conference Withdrawn Submission_

### Official Review · Reviewer_iPdY · 2025-10-15

**Soundness:** 2
**Presentation:** 1
**Contribution:** 1
**Rating:** 0
**Confidence:** 4

**Summary:**

The paper proposes a continual anomaly detection framework built on a class-conditional diffusion model with generative replay. Instead of storing raw exemplars, a frozen teacher model is used to regenerate past normal samples after each task, and a Weibull-based filter selects only “high-quality” generations for replay. A fixed projection layer defines a latent space where per-class EVT (Weibull) distributions are fitted and later used at inference time to classify test samples as normal or anomalous. Experiments are conducted on MVTec-AD in an object-incremental setup, reporting image-level AUPR, accuracy, and F1 across experiences.

**Strengths:**

- The paper addresses a relatively under-explored problem setting, continual anomaly detection with generative replay, which is still emerging.
- The proposed pipeline is simple and modular: a class-conditional diffusion U-Net with an auxiliary classifier, a frozen projector with whitening to define a stable embedding space, and per-class EVT fits used for scoring. This makes the method easy to implement and ablate. The same EVT machinery is used for two roles: gating generative replay samples and producing class-specific anomaly scores at test time, which is coherent.

**Weaknesses:**

- The opening sentence frames the problem as one of temporal drift and distributional evolution within a single data stream (for example, tooling changes, sensor drift, or evolving definitions of “normal”), suggesting a continuous adaptation scenario. However, the actual formulation and experiments correspond to object-incremental learning, where each MVTec category is a distinct, static domain introduced sequentially. This disconnect makes the framing misleading and weakens the motivation of the proposed approach.
- The proposed method lacks motivation for the use of EVT/Weibull modeling. The approach seems heuristic.
- The related work section is underdeveloped and omits several key papers on continual anomaly detection and generative replay. Recent CAD frameworks on MVTec/VisA,such as UCAD [1], as well as more recent pixel-level continual anomaly detection studies [2], and IUF [3], should be acknowledged, as they adopt the same object-incremental setting. Likewise, diffusion-based replay approaches like ReplayCAD [4] and CDAD [5] are highly relevant yet unmentioned. Without engaging with these works, the paper’s positioning and baseline choices are difficult to evaluate.
- The evaluation protocol is incomplete. The paper only reports image-level F1 / AUCPR / Accuracy, whereas pixel-wise localization metrics such as PRO / AUPRO / pixel-wise AUROC are standard for anomaly detection (especially on MVTec). Without pixel-level results, it is unclear whether the model is actually detecting where anomalies occur or just performing binary classification.
- The proposed method underperforms compared to most baselines reported in the paper, calling into question the claimed advantages of the approach and the effectiveness of the proposed components.
- The paper lacks substance and polish in presentation. Many sections are very short, some figure have undescriptive captions, and some tables are missing descriptive titles or commentary. Some pages contain only a few small plots with large unused whitespace, and the submission does not come close to the page limit. While length alone is not a criterion, the overall layout suggests underdeveloped analysis, with missing discussion, qualitative inspection, or deeper ablations that would normally be expected in a full ICLR submission.

**References**

[1] Jiaqi Liu et al., “Unsupervised Continual Anomaly Detection with Contrastively-learned Prompt,” AAAI 2024.

[2] Nikola Bugarin et al., “A Benchmark for Pixel-Level Anomaly Detection in Continual Learning,” CVPR Workshops 2024.

[3] Jiaming Tang et al., “An Incremental Unified Framework for Small Defect Detection in Industrial Inspection,” ECCV 2024.

[4] Lei Hu et al., “ReplayCAD: Generative Diffusion Replay for Continual Anomaly Detection,” IJCAI 2025.

[5] Xiaofan Li et al., “One-for-More: Continual Diffusion Model for Anomaly Detection,” CVPR 2025.

**Questions:**

- Why is quality-gated replay needed if training data are normal? If the real training images are clean, gating only makes sense if teacher-generated replay is sometimes off-manifold. Please justify.
- What is the quality of the reconstructed images? Why not use them instead of doing a projection on a U-Net layer?
- What hyperparameters were used? What are the weights used for each component of the loss?

---

### Official Review · Reviewer_wtpC · 2025-10-25

**Soundness:** 1
**Presentation:** 1
**Contribution:** 1
**Rating:** 2
**Confidence:** 4

**Summary:**

This paper proposes DiffCont, a continual anomaly detection framework designed for industrial inspection tasks where new classes or defect types arrive over time. The key idea is to combine diffusion-based generative replay with Weibull-based outlier rejection.

**Strengths:**

The method is efficient and realistic for industrial use cases.

**Weaknesses:**

1. The paper is poorly written and far from the standard of a publication. The overall writing, formatting, and organization are confusing, figures and tables are misaligned, captions are unclear, and many sentences are grammatically incorrect or lack precision. These issues make the paper difficult to read and significantly reduce its credibility.
2. The experimental section is confusing and lacks substance. Only a single dataset (MVTec-AD) is used, and the setup of “Experiment 1/2/3” is confusing. Moreover, the proposed method performs worse than baselines with a large gap. And no analysis on the experimental results. Overall, the experimental validation is incomplete and fails to support the claimed contributions.
3. The proposed method merely stitches together existing components diffusion-based replay and Weibull tail modeling without any genuine algorithmic innovation or theoretical insight. There is no deeper analysis of why the combination works, nor any exploration beyond this straightforward integration. Given that the results are also weak, the overall contribution is limited.

Overall, the paper is significantly below the expected standard for conference publication.

**Questions:**

See weakness.

---

### Official Review · Reviewer_Eak6 · 2025-11-01

**Soundness:** 2
**Presentation:** 1
**Contribution:** 2
**Rating:** 0
**Confidence:** 4

**Summary:**

This paper tackles the problem of continual anomaly detection in a class-incremental setting, motivated by industrial inspection scenarios where new object categories are introduced over time. The authors propose DiffCont, a method centered on a class-conditional diffusion U-Net. The method is evaluated on a challenging 15 class-incremental benchmark derived from the MVTec-AD dataset.

**Strengths:**

1. The paper addresses the continual anomaly detection in a class-incremental setting, which is a challenging and practical scenario.

**Weaknesses:**

1. The paper looks like a report, not a paper.

2. No analysis on any experiments, only tables

**Questions:**

1.  Could the authors clarify the experimental setup for the baselines in Table 2? Were models like PatchCore and CFA trained in the same class-incremental setting? If so, how were they adapted? If not, why are they considered appropriate comparisons for a continual learning method?

2.  The method relies on at least 5 loss-weighting hyperparameters. How were these selected, and how sensitive is the model's performance to these choices?

---

### Official Review · Reviewer_Pa7i · 2025-11-01

**Soundness:** 2
**Presentation:** 1
**Contribution:** 2
**Rating:** 2
**Confidence:** 4

**Summary:**

This paper proposes DiffCont, a continual anomaly detection framework that integrates diffusion models, outlier rejection via Weibull modeling, and generative replay for industrial inspection tasks. The approach targets the challenge of adapting to new data distributions, e.g. new defect categories while preventing catastrophic forgetting. Experiments are conducted on the MVTec-AD dataset in a class-incremental setting (15 sequential tasks).

**Strengths:**

-   The paper effectively combines diffusion-based generative replay with EVT-based (Weibull) outlier rejection, bridging continual learning, anomaly detection, and open-set recognition.

- The use of Weibull tail modeling is theoretically grounded in Extreme Value Theory (EVT). This gives a principled approach to sample acceptance and anomaly scoring.

**Weaknesses:**

- Although the paper presents baseline results, the comparison is mainly restricted to *static anomaly detection* methods. It lacks experiments against continual-learning-based anomaly detection or generative replay baselines. Moreover, the comparison does not include more recent methods from the past one or two years. The experimental evaluation is also limited to MVTec-AD. Datasets such as **RealIAD** or **MVTec 3D** are not explored. In Section 3.3 (Generation Memory Budget), the claimed advantages are not supported by experiments or quantitative evidence. The explanations accompanying tables and figures are insufficient to clearly demonstrate the effectiveness of the proposed approach. Overall, the experimental scope and baselines are insufficient to support the claims.

- Only two simple ablations are presented (number of epochs and tail size). More insightful analyses such as removing the outlier rejection module, freezing or unfreezing the projector, changing replay sample quality, or disabling the distillation loss would provide a deeper understanding of the causal contributions of each component. It remains unclear why the ablations are not conducted across multiple modules or losses.

- The overall writing and presentation are poor. There are many broken sentences and unfinished paragrahs. It feels like the paper is not ready for submission yet.

**Questions:**

- The authors should further improve the writing and presentation before submission.

---

### Note · Authors · 2025-11-20

**Comment:**

We would like to formally withdraw our submission DiffCont (Submission 18770) from further consideration.

We sincerely thank all reviewers (Pa7i, Eak6, wtpC, and iPdY) for their detailed and constructive feedback. The reviews consistently highlight several substantial shortcomings of the current manuscript, including:

- Poor writing quality, organization, and formatting, which makes the paper difficult to follow and clearly below the expected standard for conference publication.

- An insufficient experimental evaluation, limited to MVTec-AD, with unclear setups (e.g., Experiment 1/2/3) and missing analysis and discussion of the results.

- Lack of comparisons to more relevant and recent continual anomaly detection and diffusion-based replay methods, as well as missing pixel-level metrics that are standard for industrial anomaly detection benchmarks.

- Inadequate ablation studies and missing justification or analysis for key design choices, including EVT/Weibull modeling, replay gating, and hyperparameter selection.

- The fact that the proposed method underperforms several baselines, which weakens our claims and indicates that the approach and its evaluation require more fundamental rethinking.

Given the extent of these issues, we recognise that the current version of the paper is not ready for publication and cannot be fixed with minor revisions within the current review cycle. We therefore prefer to withdraw the submission and undertake a substantial reworking of the method, experiments, and presentation before considering a future submission.

We are grateful to the reviewers and the program committee for their time and feedback, which will be very valuable in guiding a thorough revision of this work.

**Withdrawal Confirmation:**

I have read and agree with the venue's withdrawal policy on behalf of myself and my co-authors.